# A Review of Lunar Communications and Antennas: Assessing Performance in the Context of Propagation and Radiation

**DOI:** 10.3390/s23249832

**Published:** 2023-12-14

**Authors:** Elham Serria, Rida Gadhafi, Sara AlMaeeni, Husameldin Mukhtar, Abigail Copiaco, Raed Abd-Alhameed, Frederic Lemieux, Wathiq Mansoor

**Affiliations:** 1College of Engineering and IT, University of Dubai, Dubai 14143, United Arab Emirates; rgadhafi@ud.ac.ae (R.G.); hhadam@ud.ac.ae (H.M.); acopiaco@ud.ac.ae (A.C.); wmansoor@ud.ac.ae (W.M.); 2Mohammed Bin Rashid Space Centre, Space Robotics Laboratory, Dubai 211833, United Arab Emirates; sara.almaeeni@mbrsc.ae; 3Faculty of Engineering and Informatics, University of Bradford, Bradford BD7 IDP, UK; r.a.a.abd@bradford.ac.uk; 4School of Continuing Studies, Georgetown University, Washington, DC 20001, USA; frederic.lemieux@georgetown.edu

**Keywords:** lunar communication, sleeve-balanced antenna, lunar antennas, lunar surface propagation, lunar propagation channels, Rashid rover

## Abstract

Over the previous two decades, a notable array of space exploration missions have been initiated with the primary aim of facilitating the return of both humans and robots from Earth to the moon. The significance of these endeavors cannot be emphasized enough as numerous entities, both public and private, from across the globe have invested substantial resources into this pursuit. Researchers have committed their efforts to addressing the challenges linked to lunar communication. Even with all of these efforts, only a few of the many suggested designs for communication and antennas on the moon have been evaluated and compared. These designs have also not been shared with the scientific community. To bridge this gap in the existing body of knowledge, this paper conducts a thorough review of lunar surface communication and the diverse antenna designs employed in lunar communication systems. This paper provides a summary of the findings presented in lunar surface communication research while also outlining the assorted challenges that impact lunar communication. Apart from various antenna designs reported in this field, based on their intended usage, two additional classifications are introduced: (a) mission-based antennas—utilized in actual lunar missions—and (b) research-based antennas—employed solely for research purposes. Given the critical need to comprehend and predict lunar conditions and antenna behaviors within those conditions, this review holds immense significance. Its relevance is particularly pronounced in light of the numerous upcoming lunar missions that have been announced.

## 1. Introduction

In October 1958, the National Aeronautics and Space Administration (NASA) commenced its operations and started progress toward the landing of a human on the moon in the 1960s through a project called Apollo, which involved astronauts orbiting the moon and landing on its surface between 1968 and 1972 [1]. When Neil Armstrong and Buzz Aldrin, the Apollo 11 astronauts, took their lunar walk on 20 July 1969, it became one of the most significant events in the history of the world [2,3,4]. In the year 2004, President Bush entrusted NASA with the responsibility of overseeing space exploration efforts and being in control of the following: 1. The return of humans to the moon. 2. The development of advanced knowledge, technologies, and infrastructures. 3. Enabling human presence across the solar system. 4. Encouraging the participation of international and commercial organizations in this project. 5. Initiating missions involving humans and robotics to the moon [5,6]. NASA conducted comprehensive architecture studies to shape lunar mission strategies. The Lunar Architecture Team had two phases of research in 2006–2007 to outline lunar surface systems and infrastructure plans. Simultaneously, NASA’s Science Mission Directorate initiated a lunar science robotic mission strategy. The inaugural undertaking involved a compact orbiter focused on atmospheric and dust science, scheduled for a 2011 launch. Subsequent missions aimed to deploy mini-landers for geophysical studies, with the initial pair planned for a 2014 launch [7]. Afterward, America started on a groundbreaking journey in space exploration, science, and technology, poised to delve deeper into the moon’s mysteries than ever. This ambitious endeavor is known as the Artemis program, named after Greek god Apollo’s twin sister, the moon goddess. Launched in 2017, Artemis entails a series of continuous lunar missions under NASA’s guidance, aiming to comprehensively uncover the moon’s secrets through a combination of scientific inquiry and human exploration objectives [8,9,10]. China launched the Chang’e-5 (CE-5) lunar exploration mission in 2020. The Lunar Regolith Penetrating Radar (LRPR) is a high-resolution imaging radar installed on the CE-5 lander to probe the lunar regolith’s thickness and fine structure in the landing region. The radar’s purpose Is to investigate the electromagnetic properties, mineral content, structure, creation, and development of the lunar regolith. Characterizing the lunar regolith with the LRPR can aid drilling and sampling operations [11,12]. Very recently India achieved a monumental feat by executing a historic landing at the lunar south pole as part of the Chandrayaan-3 mission [13]. The Vikram lander touched down on the moon with the primary objective of locating water-based ice, crucial for future human habitation and interplanetary missions. Equipped with five scientific instruments, it aims to analyze the lunar surface, atmosphere, and subsurface tectonic activity. There is also a burgeoning interest in lunar communication systems to facilitate data transfer among lunar assets and establish reliable communication with Earth stations [14]. Among these, the most recent endeavor is the Emirates Lunar Mission (ELM), an initiative spearheaded by the Mohammed Bin Rashid Space Centre (MBRSC) [15]. This ambitious project aims to conceive and actualize the UAE’s inaugural robotic lunar mission, with the Rashid rover playing a pivotal role in this groundbreaking venture. The Rashid rover was launched in December 2022 from Cape Canaveral Space Force Station in Florida, marking a historic moment in the UAE’s space exploration journey. Regrettably, during its landing, the lander could not achieve a soft landing, consequently preventing it from carrying out its intended operations. Nevertheless, the Rashid rover is part of a series of rovers that MBRSC plans to send for exploration on the moon and other celestial bodies in the future. The Rashid rover is outfitted with advanced cameras—a high-resolution camera for detailed imagery, a microscopic camera for capturing fine particulars, and a thermal imaging camera. Additionally, the rover is equipped with a specialized Langmuir probe, intended for delving into the mysteries of lunar plasma and unraveling the enigma behind the moon’s notably adhesive dust. Its primary mission was to investigate the lunar surface, assess mobility on the moon’s terrain, and explore the interplay between diverse surface materials and lunar particles. Lunar missions have played a pivotal role in advancing our understanding of the moon and its surroundings, facilitating groundbreaking scientific research and paving the way for future lunar expeditions, including crewed missions. These missions have relied on a complex network of communication to ensure the successful exchange of critical data, instructions, and insights between Earth and lunar mission components. The significance of lunar communication becomes evident when we consider the multifaceted nature of lunar missions. These missions have encompassed a wide range of objectives, from exploring the lunar surface’s geological and topographical features to conducting experiments and observations that contribute to our knowledge of the moon’s history and potential resources. Moreover, lunar missions have acted as precursors to human exploration, making the establishment of reliable communication pathways of paramount importance. In this context, various communication pathways have been established, each tailored to the specific needs of lunar missions. These pathways include Earth-to-lunar relay satellites, which serve as essential intermediaries for transmitting signals between Earth and lunar mission components. Additionally, direct communication links between Earth and lunar landers or the lunar surface itself have been vital for data transfer and real-time operations. Furthermore, lunar surface-to-lunar surface communication has played a pivotal role in coordinating activities, sharing scientific findings, and enhancing mission efficiency. This paper explores the challenges and intricacies of lunar communication, aiming to provide a comprehensive understanding of how these communication systems operate in the lunar environment. By examining the unique characteristics of lunar propagation and channel modeling, we aim to shed light on the solutions and advancements required to ensure the success of lunar missions and expeditions. In light of these missions, it becomes evident that humans need to have a clear and accurate understanding of what is happening on and around the moon. The moon is completely unlike Earth; it has lighter and darker areas on its terrains (land) and lunar maria (seas), respectively. Even though we know the moon had no water, the dark areas are still called maria. Scientists need to answer many important questions about the moon [16]. The moon undergoes extreme temperature fluctuations and is covered by regolith, with a deeper layer called “mega-regolith.” Lava tubes are considered for protection and storage, but a persistent dust cloud and magnetic field variations pose challenges to lunar communication. These anticipated atypical responses of electromagnetic waves in lunar environments can result from various factors, giving rise to phenomena such as signal reflection, diffraction, attenuation, and polarization. For instance, signal reflection off the lunar surface can introduce delays and interference in communication, making non-line-of-sight communication a challenging task. The absence of a lunar atmosphere can lead to signal attenuation, which poses particular problems for long-range communication. Furthermore, multipath propagation resulting from the irregular lunar terrain further complicates signal reception. These effects have the potential to significantly impact the performance of communication systems, especially those designed for non-line-of-sight communication, coping with multipath propagation, or operating at low data rates. Additionally, the extreme temperature conditions on the lunar surface can contribute to the degradation of communication system components. Antennas are crucial for maintaining signal quality and robust data transmission in this complex lunar environment.

This article provides an in-depth examination of the challenges encountered in lunar communication from the reported data and the profound implications they carry for establishing successful connections. The primary part of the article compares and summarizes various propagation models developed and reported in the field of lunar communication. The secondary part of the article focuses specifically on lunar antennas, delving into their design intricacies in search of an optimal solution that can facilitate precise data transfer with minimal signal losses across the moon’s surface and during transmissions between the lunar and Earth realms. Through a comprehensive analysis, this article scrutinizes a plethora of designs that have been utilized in practical missions and scholarly investigations alike. It also reviews the sleeve dipole antenna used in the Rashid rover, ambitiously deployed to explore the Atlas Crater [15]. A notable highlight of this sleeve dipole antenna is that it serves as the primary communication apparatus for the Rashid rover [17]. Additionally, this article reviews the patch antennas that act as a secondary communication channel for the Rashid rover [18]. This holistic approach ensures a thorough understanding of the communication systems’ interplay and their resilience in the lunar setting and we believe that our findings will be interesting to the scientific community. As a summary, in addition to providing a thorough examination of lunar communication systems and the antennas employed for this purpose, this article delves into the anticipated hurdles of lunar communication. The rest of this article follows the following structure:

Section 2 reviews the lunar surface propagation models reported so far. It also details various types of signal losses reported thus far. In Section 3, the paper outlines the numerous challenges posed by the lunar environment and their implications for successful communication. The primary focus of this paper centers on various antenna types and designs for lunar communication, as detailed in Section 4. Within this context, the paper provides an in-depth analysis of multiple designs reported to date. These designs encompass the Compact Microstrip Monopole Antenna (CMMA), Folded Hilbert Curve Fractal Antenna (fHCFA), Vivaldi antenna, all-metal circular patch antennas for the Rashid rover, Bow-Tie antenna, inf-IRA antennas, all-metal reflector antennas, polyimide film antennas, and dipole and monopole antennas, including the sleeve-balanced antenna used on the Rashid rover. These diverse lunar antenna designs are well-segregated into four distinct categories: planar antennas, reflector antennas, antennas for low-frequency radio astronomy, and wire antennas. This categorization stems from an extensive literature review conducted. Section 5 provides a concise overview of lunar communication and lunar antennas along with potential directions for future research. The major conclusions are outlined in Section 6.

## 2. Lunar Surface Propagation Path Loss Models

To enhance surface exploration, multi-robot systems are being extensively researched. Accordingly, understanding the operational environment is crucial for accurate radio propagation modeling due to differences between lunar and terrestrial conditions. While propagation models for signal strength are well-established on Earth, their suitability for lunar sites requires careful evaluation due to the unique lunar conditions [19]. Different approaches to propagation modeling can be seen in the literature. The very well-known approach is that of NASA [20], which presented the application of an adapted Longley–Rice model for irregular terrains along with digital elevation data that accurately represent a real lunar location. This model is well suited to estimating path loss deviation from theoretical attenuation over a reflecting surface. This model also validated the field data collected during the Apollo project. This approach also enables the estimation of radio frequency (RF) path loss across the lunar landscape. Another approach was to introduce lunar propagation models that take inspiration from various established terrestrial communication models such as Fresnel zones, the free-space path loss model, and the irregular terrain model including reflection and diffraction due to irregular terrains [19]. This approach was used to formulate a radio propagation model tailored to the lunar environment, specifically designed for micro-rovers equipped with low-height antennas. By utilizing this newly developed model alongside the Digital Elevation Model of the Apollo 15 Mission’s landing site, a series of simulations were conducted to forecast the feasible communication range and the quality of links for three different micro-rovers. Another approach is to utilize the free-space model, two-ray model, and spherical diffraction model to estimate radio attenuation on the lunar surface [21]. This model can accurately predict the link budget in lunar surface communication. The results were close to that of the model predicted in [20]. In [22], a solution for predicting path loss in lunar communication when encountering obstacles is outlined. This method simplifies transmission loss across the lunar surface into four typical scenarios based on lunar surface irregularities and obstacle shapes. The analysis accounts for free-space transmission loss, ground proximity loss, diffraction loss, and antenna gain. Additionally, data from the Apollo project were incorporated into this study, demonstrating that their accuracies align well with the model presented in [20]. Moreover, a lunar wireless model was developed in [23]. In this proposed model, the Digital Elevation Model from four distinct lunar sites is employed. These models were generated through measurements taken by the Terrain Mapping Camera, which was part of Chandrayaan-1, a recent lunar mission conducted by India. A lunar wireless mode examines all potential occurrences and signal degradations relevant to the wireless sensor network deployed on the lunar surface and anticipates minimal diffraction loss compared to the primary factors of direct path attenuation and multipath signal reflections for the specific intended usage. Given the primary objective of establishing unobstructed line-of-sight communication between transmitting and receiving units, this level of loss was disregarded. In [24], there was an attempt to develop a propagation analysis incorporating terrain characteristics, frequency considerations, and antenna height. The aforementioned models did not directly account for these factors. Given that reflections and diffractions are the dominant mechanisms in terrain-specific effects at higher frequencies, it becomes crucial to realistically incorporate these effects. In this context, the geometrical theory of diffraction (GTD) was employed, allowing for the consideration of reflections and diffractions from three-dimensional (3-D) lunar terrain, including craters. The study yielded promising results, highlighting the impact of antenna heights, frequency, and lunar ground material on antenna performance and path loss. Notably, it was observed that path loss increased with higher frequencies. There is an ongoing need for improved path loss models, especially at higher frequencies. This area of research remains an open question, with numerous researchers striving to introduce innovative approaches for crafting a more realistic model that encompasses the real terrain effects found in the lunar environment. Table 1 summarizes the propagation models reported so far.

## 3. Lunar Communication Challenges

Scientists encounter a wide range of challenges that demand thorough study and analysis to achieve clear communication without any loss of information (see Figure 1).

The moon displays a wide temperature range on its surface. Daytime near the equator can hit 250 °F (120 °C, 400 K), nights drop to −208 °F (−130 °C, 140 K), and a record-breaking −410 °F (−250 °C, 25 K) was detected at Hermite Crater, making it the solar system’s coldest known spot. Thermal investigations indicate that the temperature at Atlas Crater has the potential to exceed 80 °C during the midday period on the moon. Similarly, frigid areas were located in shadowed craters near the southern pole during winter nights [25,26,27]. Furthermore, small impact-created craters are crucial on airless solar system bodies like the moon, which accumulate due to the lack of weathering. Yet, erosion from micrometeorite impacts affects smaller craters [28]. Manual crater detection via visual inspection is impractical due to the moon’s numerous small craters. This results in limited databases with either wide coverage of large craters or coverage of craters of varied sizes in a specific area. Expert counting also shows up to 40% of disagreement [28]. Six decades of lunar exploration have yielded only 1675 dated craters out of 9137 recognized. The struggle lies in identifying diverse craters and estimating their ages, complicated by intricate morphologies [29]. Crater environments cause significant signal reflection, resulting in multiple signal copies with different strengths and delays due to diffraction and reflection. These signals include scattered energy, reflections, and direct and diffracted paths, forming a complex channel with attenuation, delay variations, and polarization. The moon’s numerous distant craters intensify signal delays, posing challenges for high-data-rate wireless systems. Empirical studies highlight multipath propagation as the primary constraint rather than weak signal strength [24].

The moon’s surface is covered by a substantial layer of regolith (fragmented and loose rock material). This regolith results from meteoroid impacts and solar particle bombardment, with varying thicknesses (4–5 m in mare regions, 10–15 m in highlands) and sizes, ranging from boulders to dust particles. While “lunar soil” and “regolith” are often used interchangeably, “lunar soil” specifically means the subcentimeter part, but, practically, it usually refers to the submillimeter fraction [30,31]. The term “lunar soil” is misleading as the moon’s soil differs significantly from Earth’s soil. Lunar soil lacks organic matter and forms mechanically due to meteoroid impact and solar–wind interaction. Unlike terrestrial soils shaped by wind and water, lunar soils are unsorted and characterized by sharp, fresh surfaces. On the other hand, lunar regolith consists of rock fragments, mineral chips, impact glasses, volcanic materials, and a unique moon-specific element known as “agglutinates” [30]. Lunar regolith displays varying permittivity and conductivity characteristics depending on its location. These characteristics directly affect the radiation pattern of antennas. Ground reflections introduce multiple peaks and valleys in antenna patterns, leading to signal oscillations. Specifically, strong reflections from the conductive lunar surface result in deep nulls in signal reception [24].

Lava tubes form when drained lava solidifies on the ceiling, providing robust protection and storage potential due to their sturdy basaltic roof. Repurposing them for storage is feasible due to their size. Resources from nearby volcanic eruptions containing valuable elements could be utilized for construction. Extracting resources from the tunnels could also be explored through drilling and expanding the existing network. Lava tunnels, with stable temperatures around 250 K, are potential deployment sites, despite challenging technology requirements [32,33,34]. Effective line-of-sight communication within lava tubes is impeded by the substantial pit depth and shadowed areas along the walls. In this context, the free-space model is inapplicable due to the significant impact of reflections and diffractions on received signal strength, resulting in substantial propagation losses [35].

Researchers discovered a persistent dust cloud around the moon from fast-moving comet particles, contrasting with slower asteroid impacts. This occurs during meteor showers like the Geminids and could extend to other airless celestial bodies [36,37]. The moon’s dust cloud can affect equipment function, necessitating careful instrument design to protect delicate components from lunar conditions [27]. Sand and dust have a more pronounced impact at higher frequencies, especially at millimeter-wave frequencies. Their effects include signal backscattering, interference related to received signal time delays, shifts in signal polarization, and the absorption of transmitted signal energy [38]. The moon’s magnetic field variations stem from rocks with permanent magnetization formed during its strong core field dynamo phase, also indicating changes in the moon’s rotational axis during the era of the active core magnetic field dynamo [39]. The magnetic field is roughly one thousand times lower than on Earth [27]. The moon’s surface becomes charged with an electrostatic potential that seeks to minimize the total incoming current [40]. Four primary sources contribute to charging currents on the moon, shown in Figure 2: photoemission of electrons, plasma electrons, plasma ions, and secondary electrons. Photoemission is the main daytime source, leading to a positive charge on the dayside, while plasma electrons dominate at night, causing the lunar surface to become negatively charged. Lunar surface charging processes are still inadequately comprehended, carrying uncertainties about potential variations in space and time [41]. A denser plasma can cause radio waves to scatter. The most recent Indian lunar mission Chandrayaan-3 revealed that the lunar surface plasma is sparse, allowing radio waves to easily penetrate it.

For a considerable duration, space scientists have extensively examined the challenges of lunar communication, recognizing their substantial impact on wireless communication such as harsh temperatures between 100 °C and −173 °C, depending on the landing sites, intensive solar radiation, the interference of lunar dust with hardware, long-distance communications with Earth stations, the importance of line-of-sight (LOS) requirements, signal propagation delays, and antenna orientation to keep a strong signal connection, including the data rate and power transmission that must comply with international regulations. These challenges possess the capability to both enhance and degrade the quality of signal transmission, in which the SNR (signal-to-noise ratio) values in lunar communications can vary significantly based on specific mission parameters, hardware, frequencies used, and environmental conditions. The SNR values for a particular lunar communication link can be determined through link budget analysis, which considers the full range of system parameters and environmental factors to ensure reliable communication in lunar missions [42].

## 4. Lunar Antenna Types

Within this section, we offer a comprehensive overview of the diverse types of antennas employed in lunar applications. Our diligent efforts to conduct a thorough literature review are evident, despite the challenges posed by the limited availability of articles concerning lunar antennas. These antennas are classified into planar antennas, reflector antennas, low-frequency antennas, and wire antennas. Additionally, considering their applications, these antennas are further divided into two sub-categories: mission-oriented antennas, utilized in active lunar missions, and research-focused antennas, exclusively employed for academic investigations. Our objective is to furnish a comprehensive synthesis of the encountered antennas, organized by their respective types. Antennas designed for specific missions should offer advantages such as a compact profile, easy installation, low power requirements, and durability, all while being shielded by a protective radome. Figure 3 shows this summary.

### 4.1. Planar Antennas

In recent decades, planar antennas have gained significant attention in lunar/space research. These antennas offer several advantages, making them an attractive option for a variety of applications. One of their primary advantages is their cost-effectiveness compared to waveguide-based antenna technology as well as their lightweight and compact design, which is crucial for many commercial applications like those for handset antennas or base stations. Moreover, planar antennas offer an advantageous platform for deploying expansive arrays, facilitating the seamless integration of supplementary electronics like amplifiers and phase shifters. These electronic modules play a pivotal role in various critical areas, including electronic warfare tactics, satellite communication systems, millimeter-wave imaging technologies, and radar functionalities. Furthermore, planar antennas are highly advantageous for various applications due to their flat shape. Specifically, they are ideal for situations where shape and size are crucial, such as using conformal printed antennas on aircraft fuselages. The ease of incorporating microwave or millimeter-wave circuit elements renders planar antennas a favored option among both designers and scientists for utilization in lunar and space communication applications. Among the planar antennas, the most common types used in lunar communication are patch and resonant slot antennas due to their exceptional parameters with miniature form factors [43].

#### 4.1.1. Patch Antennas

NASA’s Glenn Research Center (GRC) has investigated numerous miniature antenna designs that could be employed for communicating on the surfaces of planets and moons. Among the designs explored, two have shown potential: the CMMA and the fHCFA. Both of these antennas were designed and tested in the research laboratory. While both of these antennas incorporate certain non-planar elements, such as the vertical metallic enclosure in the case of the CMAA antenna and the third-order Hilbert curve that folds in upon itself into four layers incorporated into the fHCFA antenna, their fundamental structure still relies on a patch and microstrip configuration. Therefore, these antennas have been classified within the planar category despite these non-planar components. GRC has developed and simulated the CMMA for lunar and space communication applications using IE3D simulation software (http://www.zeland.com/) (accessed on 4 October 2023). The CMMA exhibits a unique configuration characterized by a three-lobed patch enclosed by a vertical metallic barrier and a grounding boundary near the input location, as illustrated in Figure 4a,b. The selected substrate for the antenna is Duroid 5880, which has a thickness of 1.57 mm and a relative permittivity (ε_r_) of 2.2.

This antenna boasts an exceptionally compact form factor, measuring 12 mm × 12 mm with a height of 11 mm, making it one of the smallest antennas in the documented group. Its size corresponds to λ/12, where λ represents the wavelength in free space at its minimum operational frequency, rendering it well-suited for integration into small devices. The CMMA antenna operates at a frequency of 2.05 GHz and boasts the widest fractional bandwidth among the three reported patch antennas with a remarkable 6.3%.

Notably, this antenna delivers an impressive directivity of 6 dBi, surpassing the performance of the other two patch antennas in the study. This enhanced gain, coupled with its extensive bandwidth and directivity, positions the CMMA antenna as a promising candidate for various applications. In contrast to conventional small antennas, which are often omnidirectional, the CMMA’s attributes make it suitable for deployment in a wide range of scenarios where tiny, high-performance antennas are typically utilized [44].

NASA’s GRC center has also developed the fHCFA using Zeland’s IE3D electromagnetic simulator. This marks NASA’s second antenna designed for lunar communication. The fHCFA boasts the smallest dimensions within its antenna group, consisting of four layers with measurements of 5 mm × 5 mm × 4.5 mm (length × width × height), resulting in a λ/30 design. The antenna structure was etched onto Duroid 5880 with ε_r_ = 2.2 at a thickness of 0.127 mm. It was affixed to an aluminum ground plane using multiple layers of adhesive, with 1 mm thick HTP-6 foam spacers (ε_r_ = 1.07) separating the layers. A probe feed was employed to excite the antenna, as depicted in Figure 5a,b.

These antennas were designed for operation across multiple frequencies, with particular interest in two different operating frequencies: 2.3 GHz and 16.8 GHz. These frequency bands, specifically the S band and Ku band, are well-suited to lunar surface network communications and planetary exploratory missions that rely on local area networks (S band) and satellites (Ku band) for establishing communication links. Remarkably, this antenna’s multi-band capability eliminates the need for electrical or mechanical switching, allowing it to perform the tasks of two separate antennas while maintaining a reduced size. However, it is important to note that this antenna has the narrowest bandwidth within the group, measuring 0.5% at 2.3 GHz and 3% at 16 GHz. The directivity at the S band is measured as 1.6 dBi, while at the Ku band, it reaches 7.3 dBi [44,45].

#### 4.1.2. Slot Antennas

The tapered slot antenna (TSA) is a microstrip antenna with a coplanar structure and a tapered slot printed on a dielectric substrate. In support of the feeding apparatus, the tapered slot’s narrow extremity is affixed to the anterior portion of a receiver or the output phase of a transmitter, while its wider extremity either captures or emits radio waves in the end-fire orientation. The slot antenna offers a broad frequency bandwidth, a symmetrical radiation pattern, and end-fire radiation characteristics. The slot taper can take various geometrical forms, including linear taper (LT), exponential taper (ET or Vivaldi), constant width taper (CWT), and others, each suited to specific applications. An array of TSAs may be arranged parallel to one another and utilized for airborne radar, beam-shaping antennas, imaging systems, and phased arrays [46,47,48]. The Vivaldi antenna is a TSA that is recognized as the most advanced antenna for a variety of applications including wireless communications, biomedical engineering, and military applications [49].

A Vivaldi antenna was used in the Chinese mission due to its compatible design and broader frequency bandwidth for lunar communication. In 2020, China launched the Chang’e-5 mission, which includes the LRPR on its lander. The LRPR’s role is to analyze the lunar regolith’s properties and structure to support drilling and sampling activities [11,12]. To accomplish this goal, an ultra-wideband multiple-input multiple-output (MIMO) array was created, with the ability to conduct two-dimensional scans of the area beneath the array. This is achieved by toggling antennas between transmitting and receiving modes using a switch matrix. The MIMO array design encompasses 12 strategically positioned small quarter-elliptical slotted Vivaldi antennas (QESAs) mounted on the CE-5 lander. These antennas were constructed using a ceramic-filled polyimide laminate material with a relative permittivity of 3.8, loss tangent of 0.006, and thickness of 1 mm. A visual representation of the QESA can be found in Figure 6, and its detailed structure specifications are provided in Table 2 [50].

In order to attain extensive frequency compatibility for the antenna, the slot line and microstrip terminals were designed as a wideband short stub and open stub, respectively. The 12 elements are divided into three groups: A, B, and C. Groups A and B are positioned on the lander’s lower surface while Group C is located on the lander’s side, as depicted in Figure 7. Operating across a frequency range of 1.0 to 4.75 GHz, this antenna can deliver a realized gain ranging from 2 to 8 dBi. Its remarkable broadband capability enables it to achieve an impressive fractional bandwidth of 130%, marking it as the highest bandwidth reported in its class. However, lunar mission antennas face specific challenges. During lunar landing, the engine’s plume can subject the antenna to temperatures as high as 250 °C. Simultaneously, the plume stirs up charged lunar dust, which, if it adheres to the antenna elements in significant quantities, can adversely affect antenna performance. To safeguard against these conditions, a radome was employed to cover the antennas, capable of withstanding temperatures of up to 350 °C. These radomes were constructed from polyimide composites to ensure robust protection for the Vivaldi array [50,51,52,53,54].

#### 4.1.3. All-Metal Circular Patch Antenna

The Rashid rover employs a dual-antenna system for its communication needs. The first antenna, a sleeve dipole antenna, is exclusively dedicated to primary communication between the rover and the lander, which will be demonstrated in the succeeding section. In the event of a primary communication failure, the secondary communication subsystem comes into play using a circular stacked patch antenna. This secondary communication serves multiple purposes: it not only acts as a backup for primary communication but also fulfills a crucial mission objective, which is to establish direct communication with Earth during the second lunar day. Additionally, it serves as an experimental payload to evaluate the capabilities of an innovative and power-efficient lunar-to-Earth communication system. The primary component of the secondary antenna is an all-metal circular patch as shown in Figure 8, designed to operate at the uplink frequency of 2101.2 MHz and the downlink frequency of 2266 MHz, with a gain of 6 dBi [18]. This choice of antenna was due to its high gain and multi-band operation. Table 3, Table 4 and Table 5 compare the three reported planar lunar antennas in terms of application, performance, and simulation parameters.

### 4.2. Reflector Antennas

The progress of antenna theory has come from the success of space exploration. Because of the necessity to communicate across long distances as on the lunar surface, specialized antennas were required to broadcast and receive signals that had to traverse millions of kilometers. Reflector antennas have been used in a variety of applications throughout history because, among other antenna designs, they give the highest gain, largest bandwidth, and finest angular resolutions at the lowest cost. A reflector antenna’s principal function is to confine or radiate the majority of the electromagnetic energy passing through its aperture onto a focal plane or distant field for communication or energy transfer. Conic sections, parabolas, ellipses, hyperbolas, and spheres are examples of typical reflector antenna designs. Reflector antennas are employed in a variety of applications, including satellite communication, radio astronomy, radar, remote sensing, medical, and military applications [55,56].

#### 4.2.1. Inflatable Impulse-Radiating Antenna (Inf-IRA)

Figure 9 provides the concept and general schematic of the inf-IRA that can be used in lunar penetrating radar. It is designed to be highly portable and easily transportable, with the ability to be deflated and rolled up into a compact size for convenience. However, once air is pumped into the antenna, it rapidly inflates to a paraboloid shape, similar to that of a balloon.

The antenna comprises three primary components: a parabolic reflector, feed arms, and matching resistors. What distinguishes this antenna from traditional impulse-radiating antennas is that it features only a single feed arm, resulting in an input impedance of 400 Ω [52]. Consequently, the implementation of a balun becomes necessary. To facilitate the construction of a balun, it is necessary to decrease the input impedance of the antenna. In this work, the implementation of asymptotic conical dipole (ACD) feed arms is employed to reduce the input impedance from 400 Ω to around 200 Ω. Figure 10 illustrates the depiction of the traditional IRA and the intended ACD arm of the fed-IRA. To prevent energy from being reflected back into the antenna system, 100-ohm impedance matching resistors are placed between the ends of the 0.3 m diameter ACD feed arms and the reflector. It is important to note that for lower-frequency applications, proportional scaling of the antenna is necessary to achieve optimal performance, which includes increasing the antenna’s diameter accordingly [57,58,59].

When fully inflated with air, the edge of the reflector may become wrinkled, so margins are included along the rim to prevent inaccuracy. Eight stays are placed on the interior to prevent deviation from the parabolic form and they are fastened to the reflector vertically with flexible Kapton tape. The ACD feed arms are printed on a flexible thin Kapton substrate, which also serves as a stay to maintain the reflector’s parabolic shape. Once the air is removed and the Mylar surface is deflated, the thin Kapton substrate remains flat, allowing the inf-IRA to be folded up for portability. The inflated and curled-up morphologies of a constructed inf-IRA are shown in Figure 11 [57]. The antenna is simulated using FEKO.

In terms of functionality, this antenna functions as an impulse-radiating antenna and has the ability to cover frequencies ranging from 0.5 GHz to 4 GHz, enabling broadside operation that encompasses VHF and S band frequencies. Within its operational range, the antenna can achieve a gain ranging from approximately 2 to 17 dBi. For applications such as lunar penetrating radar, the need for the high-resolution detection of geological structures is evident. In this context, the advantages of utilizing ultra-wideband antennas that can operate across a wide frequency spectrum, from very low to high frequencies, are apparent. Additionally, employing high-directivity antennas is essential to minimize energy wastage. While these options necessitate the use of large antennas, it is important to note that larger antennas can escalate launch costs. In this regard, inflatable structures hold particular significance due to their compact packaging and minimal mass, offering a potential solution to address this challenge.

#### 4.2.2. All-Metal Reflector Antennas

All-metal antennas are advantageous in space because they can tolerate low temperatures, intense radiation, and vacuum settings. Another significant advantage is the elimination of all dielectric materials in the antenna structure, which prevents electrostatic charge accumulation and related noise. Moreover, a short backfire design may be simply made and is lightweight, without the need for complex deployment procedures [60]. In recognition of its high gain, high efficiency, minimal side lobes, great front-to-back ratio, small form, and sturdy construction, the short backfire antenna (SBFA), initially reported in 1965, has been widely employed in space, marine, and terrestrial applications. This antenna was used in, for example, the communication link between the NASA ground station and the Apollo spacecraft’s two moon capsules. The fundamental low-profile design of the SBFA has experienced very modest alterations over the past 50 years, consisting of a cylindrical cavity housing a feed element situated between a ground plane and a smaller sub-reflector [61].

In [60], construction is illustrated of an X-band short backfire antenna, specifically designed to meet the requirements of communication and navigation applications in the harsh environments of space and the lunar surface. Additionally, it enables direct communication with Earth from both the lunar surface and the lunar orbit space. As per Space Frequency Coordination Group (SFCG) recommendations, the bands 7190–7235 MHz and 8450–8500 MHz are suggested as targets for these links, and this design demonstrates suitable performance and tunability within these bands [62,63]. Moreover, due to its significant gain, a backfire element can be employed to replace numerous patch elements in an array antenna. Backfire antennas exhibit characteristics similar to those of large reflector antennas, such as high gain, equal E-plane and H-plane beamwidths, and low side lobes. However, they can be scaled down to smaller sizes, making them advantageous for space-constrained applications. The schematic image of the short backfire is presented in Figure 12a and the cross-sectional image of the antenna element is illustrated in Figure 12b, with the dimensions in Table 6, where λ_o_ is the free-space wavelength corresponding to the design frequency (8.425 GHz). The proposed antenna has a cylindrical primary reflector and a circular sub-reflector disc. The antenna is linearly polarized and has high gain due to its rectangular waveguide feed incorporated into the main reflector [60].

Multiple designs of the above-mentioned basic antenna construction have also been created and discussed in [60]. First, a concave reflector was constructed by tilting the back face of the primary reflector forward by 15 degrees. Secondly, hexagonal implementation was performed. When arraying, the hexagonal form allows for the best packing efficiency. These antennas produce a gain ranging from 13.4 to 15.6 dB in the cylindrical design and from 12.5 to 14.2 dB in the hexagonal design over the X-band frequencies. The −10 dB return loss bandwidth for the cylindrical main reflector is 8.92 to 9.4 GHz, while it is 8.0 to 8.42 GHz for the hexagonal main reflector [60,64,65], enabling a narrowband operation compared to inf-IRA. Table 7, Table 8 and Table 9 compare reflector lunar antennas in terms of application, performance, and simulation parameters.

### 4.3. Antennas for Low-Frequency Radio Astronomy

The lunar surface, particularly the far side, offers an attractive option for low-frequency radio astronomy, as elucidated in [66]. Since low-frequency antennas on the lunar surface typically have minimal gain and directivity, achieving the necessary angular resolution for observing individual astronomical objects requires the use of interferometers composed of multiple small, low-gain antennas, such as dipoles. Notably, these antennas have a significant capacitive component in their impedance, making fixed impedance networks unsuitable for larger bandwidths. Therefore, for low frequencies, it is strongly recommended to use physically large antennas approaching half-wavelength in size, along with wide conductor shapes, to enhance the bandwidth. Two antennas were found in this category: lunar gravitational and polyamide film (thin film) antennas. These antennas have different deployment procedures compared to other antenna types. Both of these antennas were created for the purpose of research.

#### 4.3.1. Lunar Gravitational-Wave Antennas (LGWAs)

The Lunar Gravitational-Wave Antenna (LGWA) project aims to create a lunar network to monitor gravitational wave-induced vibrations, paving the way for groundbreaking scientific exploration on the moon. By observing gravitational waves in the decihertz range, LGWA enhances our cosmic understanding and requires advanced components, including cryogenic sensors and noise-cancellation arrays, to minimize interference. Multiple stations with LGWA-grade arrays are crucial for effective noise reduction and local data analysis on the moon [27].

In [27], the authors found that LGWA has several unique characteristics compared to other detectors, which require specific data analysis techniques and give it distinct capabilities. LGWA stands out due to its evolving antenna pattern linked to the moon’s 27.3-day rotation, while its dispersed array across the lunar surface offers precise gravitational wave polarization measurements and extended gravity theory testing. With resonant bar antenna attributes up to 10 mHz, LGWA transitions into a wideband detector beyond this range. Notably, LGWA boasts unparalleled sensitivity in the 0.1–1 Hz frequency span among gravitational wave detectors, pending the emergence of new detection technologies.

#### 4.3.2. Polyimide Film Antennas

Due to the aforementioned requirements concerning the electrical and mechanical properties of low-frequency antennas, the utilization of thin conductors becomes particularly crucial. This choice not only helps in minimizing mass but is also enabled by the relatively low electrical conductivity of the lunar regolith on the far side. As a result, it becomes feasible to position antenna elements directly on the lunar surface without incurring substantial adverse effects on RF performance. This approach opens up new possibilities for efficient antenna deployment in lunar environments. In [66], the authors discussed the challenges of low-frequency antennas: inefficient signal transfer for short antennas, limited gain at frequencies below tens of MHz without large size, and the need to address mechanical properties for larger antennas. For lunar surface antennas, non-self-supporting thin conductors are suggested due to the low electrical conductivity of lunar regolith (10^−10^ mho/m).

### 4.4. Wire Antennas

Wire antennas have the ability to generate a concentrated radiation pattern. Typically, these antennas are meticulously crafted to match their length with the wavelength. They find widespread use across diverse domains, including navigation, meteorology, military operations, and security applications. Examples of wire antennas encompass dipole and monopole antennas, helical antennas, and loop antennas, each with distinct strengths and weaknesses, rendering them apt for specific applications [56].

#### 4.4.1. Dipole Antennas

Dipole antennas, also known as doublets, are widely used in various fields, including lunar communication, because of their omnidirectional properties, low mass, and low power requirements. A dipole antenna is characterized by a structure resembling a rod or metallic wire, separated by a certain distance. The most basic dipole configuration consists of straight wires or rods connected end to end. Typically, the dimensions of a dipole antenna are proportional to the wavelength, with the half-wave dipole antenna being the most prevalent design, where each rod has a length equivalent to one-quarter of the wavelength. The half-wave dipole has a radiation resistance of 73 Ω [67,68].

Dipoles were used in the radio observatory on the lunar surface (ROLSS), which is an innovative concept for a near-side low radio frequency imaging interferometric array aimed at researching particle acceleration in the sun and inner heliosphere. One of its objectives is to limit the density of the lunar ionosphere by investigating a lower radio frequency limit for solar radio emissions and determining the population of low-energy electrons in astrophysical sources. The design comprises three antenna arms arranged in a Y-configuration, with a central electronics package (CEP) situated at the center. The Y-shaped configuration allows for the creation of high dynamic range images in a short time and facilitates easy deployment. The antenna arms consist of linear strips of polyimide film, like Kapton, where 16 dipole antennas are deposited by a conductor, such as silver. Each one of the dipole antennas is a single polarization dipole. The strips can be rolled up for easy transport and deployed by unrolling. Transmission cables transport the radio signals from the scientific antennas to the CEP through the arms. The CEP holds the receivers for the dipole antennas, command and data handling hardware, and the downlink antenna, positioned outside the package [69,70].

The interferometric array was meticulously designed to operate within the frequency range of 1 MHz to 10 MHz [69,70]. To accomplish this, dipole antennas were strategically positioned on the polyimide film of each array arm. Notably, the dipole antenna operates at the higher end of the array, specifically at 10 MHz. However, at the lower end (1 MHz), the antenna becomes electrically short, resulting in a small feed point resistance and high capacitance, sharing similar characteristics with the previously mentioned low-frequency antennas. Within this frequency range, the antenna’s performance exhibits significant drawbacks, including poor efficiency, radiation resistance, and fractional bandwidth. Despite these shared characteristics, we opted not to classify these antennas within the previously mentioned low-frequency antenna category for two specific reasons. Firstly, these antennas are fundamentally dipole antennas, and, secondly, their resonant frequency is 10 MHz, which falls outside the conventional definition of the low-frequency spectrum.

The second dipole antenna that is used in lunar communication is the small-sized active dipole antenna. It was noted that studies have produced many ultra-long-wavelength antennas that are a prototype of ultra-long-wavelength radio telescopes for future moon radio observatories. In [71], the authors conducted a study on the complex geometry of an active dipole antenna that was situated above a partially conductive ground. This study involved numerical simulations of the antenna prototype as well as measurements of various parameters. The active dipole antenna is presented in Figure 13 [71]. The antenna is designed to operate at 1–70 MHz. However, other characteristics in terms of performance or structure are not available.

#### 4.4.2. Monopole Antennas

Monopole antennas are the most basic type of RF antenna, consisting of a single linear element with a length equal to a quarter wavelength. It is created by substituting one half of a dipole antenna with a ground plane at right angles to the remaining half. If the ground plane is large enough, the monopole acts precisely like a dipole, as though its reflection in the ground plane forms the dipole’s missing half. A typical monopole antenna feed is a coaxial wire with its inner conductor linked to the vertical monopole element via a hole in the ground plane and its outer conductor attached to the ground plane via a flange [72,73,74].

The monopole antenna is used in a moon mission by China. One notable application that utilizes monopole antennas is the Chang’e-3 spacecraft’s lunar penetrating radar (LPR), which is a key scientific instrument [75]. Its scientific objectives include surveying lunar regolith and detecting underlying geologic features. LPR is a type of carrier-free nanosecond imaging radar that operates in the time domain. The LPR system sends a pulse signal into the lunar subsurface, which reflects and scatters back to the radar upon encountering uneven layers, interfaces, rocks, or other objects. By analyzing the reflected and scattered signals, the thickness and distribution of the lunar regolith and the geological structure of the lunar subsurface can be determined along the rover’s path.

The LPR system has two detection channels: CH1 and CH2. CH1 operates in the 40–80 MHz range and has a depth resolution of several meters. The selection of the CH1 antennas for the lunar rover took into account various technical requirements, including the size and shape of the rover and possible installation locations. Two broadband monopole antennas grounded on the rover’s body were used for this purpose, as shown in Figure 14a. As illustrated in Figure 14b, the LPR’s CH1 transmitting and receiving antennas are installed via an extension mechanism on the two bottom edges of the lunar rover’s top board to enable installation. Once the lunar rover lands on the moon’s surface, a command signal is transmitted from the ground base to activate the release mechanisms, which deploy the antennas along a pre-determined path to the back of the rover. The antennas are then precisely pointed in the X direction for optimal communication and data transmission, as shown in Figure 14a. Since this antenna is used in real missions, the antenna is mounted inside a radome for mechanical support and thermal stability.

#### 4.4.3. Bow-Tie Antennas

The bow-tie antenna is constructed from a bi-triangular sheet of metal, with the feed located at its vertex. Bow-tie antennas fed by coaxial line, stripline, and coplanar waveguides offer several benefits as UWB antennas, including low profiles, excellent radiation efficiency, ultra-broad impedance bands, and ease of construction. Because of their benefits, they are widely employed in a range of applications, including ground penetrating radars, mobile stations, and pulse antennas [76,77]. Detailed geometry and impedance calculations are in [78].

The previous iteration of the LPR utilized a CH2 transceiver antenna consisting of a series of bow-tie antennas. CH2 operates in the 250–750 MHz range and has a depth resolution of less than 30 cm. To achieve an ultra-wideband capability, this set of antennas was equipped with resistors that absorbed the current reflected by the set’s end. Additionally, a shallow, rectangular conducting cavity was incorporated into the bow-tie antenna design to enhance radiation performance and prevent interference from the surrounding environment. The cavity was positioned behind the antennas to increase radiation directed toward the ground and improve the forward-to-backward ratio. The CH2 transceiver antenna was installed at the bottom of the lunar rover, approximately 30 cm above the lunar surface. Figure 15 provides a detailed construction schematic of the CH2 antenna. Each element has a size of 336 mm × 120 mm, with a spacing of approximately 160 mm between them. FEKO software and FDTD methods were used. A shallow, rectangular conducting cavity was used in the design to reduce interference from the surrounding environment.

#### 4.4.4. Sleeve Dipole Antennas

The final classification within the category of wire antennas is the sleeve dipole antenna. One of the main characteristics of sleeve dipole antennas is the simplicity of alignment between the transmitter and receiver antennas to ensure polarization matching. This antenna consists of a pair of radiation elements in the form of arms. The upper arm takes the shape of a coaxial copper cylinder, whereas the lower arm is composed of two components: a copper sleeve-shaped cylinder and an inner coaxial feeding line. The feeding is facilitated by a semi-rigid coaxial wire that passes through the lower arm, effectively linking the upper and lower sleeve arms [79]. The sleeve dipole design enables end-feeding of the antenna, eliminating cable and feed-point interactions that can disrupt its performance. The antenna itself is a balanced structure, making it compatible with unbalanced coaxial lines and eliminating the need for an additional balun for feeding. Furthermore, the sleeve antenna maintains consistent performance regardless of environmental conditions. These attributes render it well-suited to applications in radar, radio communication, and wireless technology.

In [17], the study delves into the analysis of the antenna performance on the Rashid rover situated within Atlas Crater. The sleeve dipole antenna was chosen for its exceptional thermal stability, which permits the application of various coatings without impacting the antenna’s parameters. Figure 16 provides an overview of the sleeve dipole antenna design implemented in the Rashid rover mission. This design permits the application of various thermal coatings while preserving optimal performance. To mitigate the impact of the moon’s extreme surface temperatures, which result from the lack of an atmosphere and may pose challenges to an antenna’s functionality and reliability, it is essential to thoroughly investigate these effects before initiating a mission. The antenna underwent simulations and optimizations within CST Microwave Studio (Version 2023), integrating it seamlessly into the Rashid rover. These simulations incorporated a lunar ground model mimicking the properties found in Atlas Crater. Accounting for the lunar regolith’s effects in electromagnetic simulations is crucial as it has been observed that this lossy ground can significantly affect antenna performance, particularly at higher frequencies. The average permittivity of lunar soil was determined to be 3, as referenced in [80,81,82,83]. Within the lunar environment, the antenna demonstrated a gain of 2.3 dB across a bandwidth ranging from 2.3 GHz to 2.7 GHz. Notably, the lunar ground, functioning as both a partial reflector and an absorber of electromagnetic waves, introduced multiple ripples to the antenna pattern due to ground reflections [84,85].

For a comprehensive comparison of various wire antennas, including their applications, performance characteristics, and simulation parameters, please refer to Table 10 and Table 11.

Based on the antennas employed in actual missions and the literature available, Table 12 provides a summary of recommended frequencies in lunar communication and their respective applications [86].

## 5. Discussion

In recent times, the moon has emerged as a focal point of technological research and development, particularly in the realms of in situ resource utilization, human exploration and habitation, and its potential role as a staging point for human missions to Mars [87]. NASA’s Artemis program is poised to establish a sustained presence on the lunar surface, ushering in an era of more extensive lunar exploration than ever before. This burgeoning lunar activity necessitates the development of advanced communication, navigation, and networking capabilities, and LUNANET stands as a prime example of meeting this demand. Notably, the United States, China, and India have already solidified their positions in lunar exploration, with several other nations, including the United Arab Emirates, declaring their intentions to embark on lunar missions in the near future. The moon, distinct from Earth, lacks an atmosphere, underscoring the importance of comprehending its surface and associated challenges. These missions serve as a source of national pride and demand substantial financial investments, emphasizing the significance of a thorough understanding of the lunar environment prior to their initiation. Moreover, lunar communication networks are not yet as established as their terrestrial counterparts. As noted in Section 2, researchers are actively engaged in the ongoing development of accurate propagation models. It is paramount to meticulously design and test the systems in such environments to ensure error-free operations, taking into account the limited optimization opportunities once the systems are deployed on the lunar surface. As elucidated in Section 3, lunar communication presents a multitude of challenges. Antennas, serving as pivotal components within the communication system, function as sensors for the entire network. Therefore, the stability of these antennas, both electrically and mechanically, is of the utmost importance. Conducting electromagnetic simulations is a fundamental step in this regard. As summarized in Section 4, lunar antennas possess many unique characteristics. Lightweight antennas with superior performance hold significant relevance in this context. Mission-specific antennas may necessitate additional coatings or special materials to withstand the extreme temperature conditions experienced on the lunar surface. The antennas discussed in the review article commonly serve dual roles, operating as both transmitters and receivers. This dual functionality is a prevalent feature in space communication systems, enabling bidirectional communication. It is vital to understand that the power capacity and data rate of antennas used in lunar missions vary significantly based on the specific mission’s objectives and capabilities. Larger lunar rovers and assets, equipped with high-power and fast communication systems, may necessitate antennas capable of handling higher power levels and faster data rates. Conversely, smaller lunar assets and rovers, constrained by size and power limitations, may opt for lower-speed and lower-power antennas. The choice of antennas is inherently linked to the mission’s unique size and capabilities. In addition, it should be noted that the power levels employed in microwave antenna applications for lunar missions can differ greatly based on the particular mission’s goals, specifications, and available technology. The following are some estimated wattage power level ranges for various lunar communication applications: 1. Low-power data communication, usually between 5 and 50 watts for lunar orbiters to Earth. Since these missions are not very far from Earth, dependable communication does not necessitate very high power levels. 2. Medium-power communication. This covers two classes. (a) Lunar landers to Earth—depending on the particular mission objectives and the distance to Earth, power levels can range from roughly 50 watts to 500 watts—and (b) lunar rovers for surface exploration—power outputs for short-range communication can be as little as 10 watts, but longer communication ranges can require up to 200 watts. 3. High-power long-range communication. Long-range, high-data-rate communications needed for lunar missions may require power levels ranging from several hundred to several kilowatts. The precise power level necessary for dependable transmission across the lunar distance is determined by mission specifications. 4. Spacecraft redundancy. Several transmitters and antennas are used in some lunar missions, and the combined power levels can be sufficient to handle the mission’s communication requirements. For instance, a 600-watt total output power can be achieved by combining transmitters with capacities of 100 and 500 watts. Please be aware that depending on the particular design and technology utilized in a lunar mission, these power level ranges may vary and are only estimates. The distance of the spacecraft from Earth, data transfer rates, redundancy requirements, and communication system performance all affect the power level selection. Future lunar missions might use varying power levels to accomplish their communication objectives as technology develops.

Based on the above and due to the limitations of the lunar environment, lunar communications, which involve delivering data between Earth and spacecraft or between different lunar missions and vehicles, have unique power and data rate issues. Depending on the objective, hardware, and technology employed, the actual power capacity and data throughput can vary. Thus, a number of factors affect the power capacity of lunar communication systems, including the power requirements of the communication equipment and the spacecraft’s power source (e.g., solar panels, nuclear, or RTGs—radioisotope thermoelectric generators). However, data rates for lunar communications are varied based on the communication’s goal, the distance between Earth and the moon, and the technology used. Such examples of these applications are as follows: 1. Telemetry and commands—these rates can range from a few kilobits per second (Kbps) to several tens of Kbps. 2. Science and imaging data—these data rates can range from several Kbps to multiple megabits per second (Mbps), depending on the complexity and volume of the data being collected. 3. Real-time communication—this includes live video real-time communication of critical events in which the data rates can reach multiple Mbps or more. 4. Laser communications—this applied emerging technology provides significantly higher data rates than traditional radio frequencies, with data rates in the tens to hundreds of gigabits per second (Gbps) or more. In summary, the data rate used for a lunar mission is determined by mission needs, available technology, and financial constraints. It is critical to match data rate requirements with available power capacity because higher data rates may necessitate more energy for transmission. On another hand, communication capabilities are projected to improve as technology progresses and additional lunar missions are planning to achieve better data rates and more efficient use of power in lunar communications. In addition, to avoid interference with other satellite systems, regulatory authorities such as the International Telecommunication Union (ITU) assign certain frequency bands and impose power limitations. These standards require that power levels be adhered to.

We firmly believe that this comprehensive review, encompassing multiple articles and their comparative analysis, will offer a valuable contribution to the lunar scientific community. Furthermore, this review serves as the foundation of our ongoing research efforts. Our research group is dedicated to exploring the intricate aspects of lunar antenna behavior in the lunar environment and developing models for the corresponding propagation channel in preparation for the upcoming Rashid rover mission. These findings will not only advance our understanding of lunar antenna systems but also play a vital role in optimizing the selection and placement of antennas for future missions, including the Emirates Lunar Mission (ELM).

## 6. Conclusions

Our review provides a thorough analysis of lunar communication, highlighting its critical importance in the context of human and robotic space exploration missions. We explore the complexities of signal propagation across the lunar surface, including environmental challenges that must be overcome for effective communication. The review evaluates various lunar surface antennas and their suitability for this challenging environment. The selection of antennas is a critical aspect of lunar communication, and it can vary significantly based on specific mission requirements. When designing antennas for lunar communication, it is essential to consider a range of factors. In general, antennas intended for lunar missions should offer several advantages, including a compact profile for efficient use of space, ease of installation, low power requirements to conserve resources, and durability to withstand the harsh lunar environment. Additionally, the use of protective radomes can help shield antennas from lunar conditions, ensuring their longevity and performance. These research findings suggest that the most used frequency bands in practical lunar surface missions include HF, UHF, S, and Ka bands. Additionally, it is worth mentioning that X and Ku bands have been explored for research purposes, presenting unique capabilities for communication between lunar orbiters and Earth applications. The choice of the appropriate frequency band should align with the specific communication requirements and objectives of the mission. Antennas used in lunar communication are primarily engineered for three key functions: lunar surface communication, Earth-to-lunar communication, and lunar penetrating radar applications. Each of these applications comes with its own set of challenges and considerations, such as addressing signal propagation, power requirements, and environmental factors. In conclusion, when designing antennas for lunar communication, it is crucial to carefully evaluate the specific needs of the mission and select the appropriate antenna type and frequency band. Additionally, ensuring compatibility with the lunar environment and mission objectives is paramount. By considering these factors, antenna designers can play a vital role in the success of lunar missions and lunar research efforts.

## Figures and Tables

**Figure 1 sensors-23-09832-f001:**
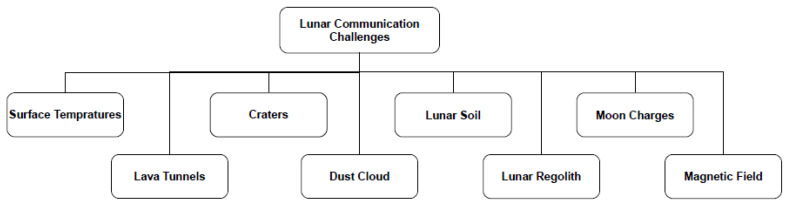
Lunar communication challenges.

**Figure 2 sensors-23-09832-f002:**
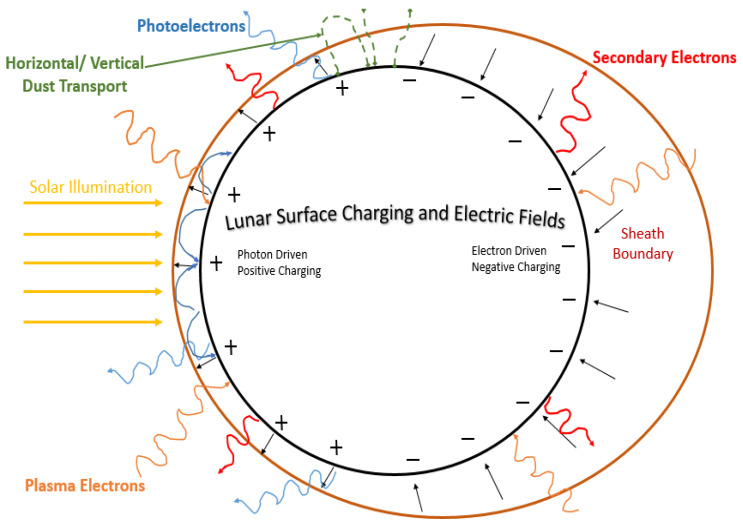
Schematic of the lunar environment [41].

**Figure 3 sensors-23-09832-f003:**
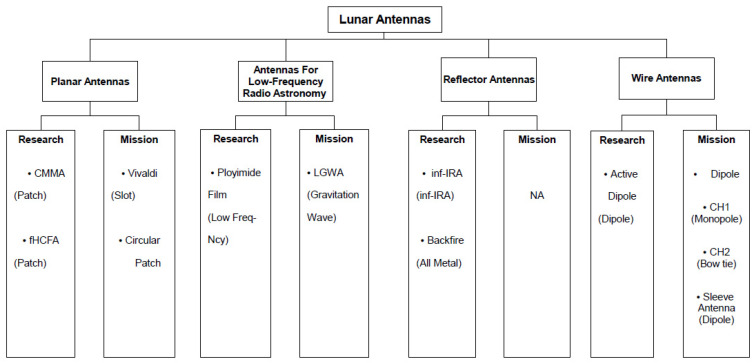
Classification of lunar antenna types based on the existing literature.

**Figure 4 sensors-23-09832-f004:**
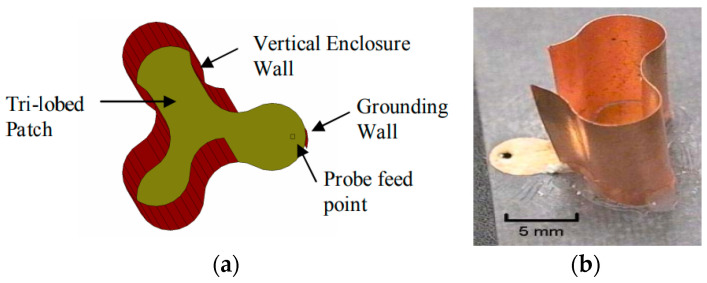
(**a**) CMMA 3-directional view. (**b**) CMMA prototype [44].

**Figure 5 sensors-23-09832-f005:**
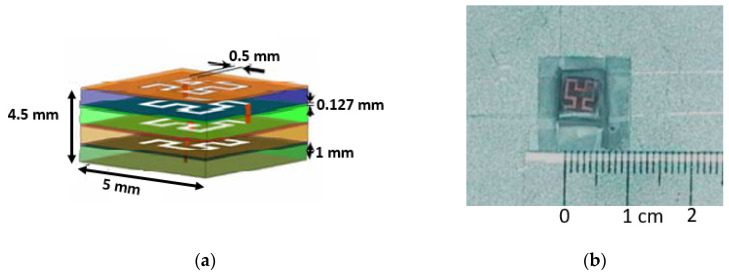
(**a**) fHCFA with dimensions layout. (**b**) Fabricated fHCFA [44].

**Figure 6 sensors-23-09832-f006:**
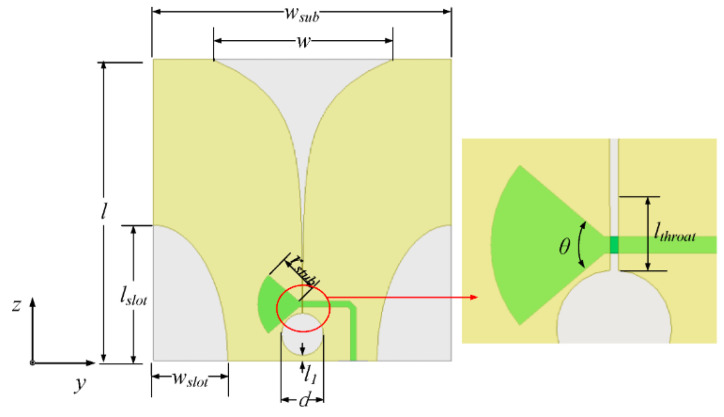
Quarter elliptical slotted antenna (QESA) [50].

**Figure 7 sensors-23-09832-f007:**
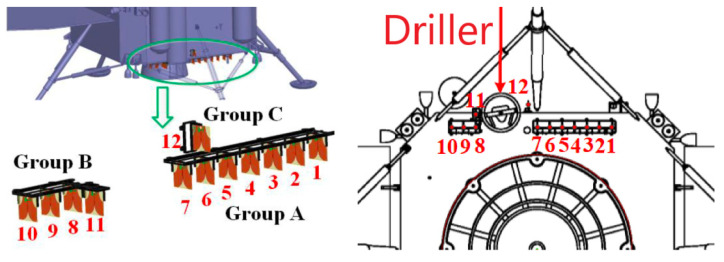
Mimo array layout [50].

**Figure 8 sensors-23-09832-f008:**
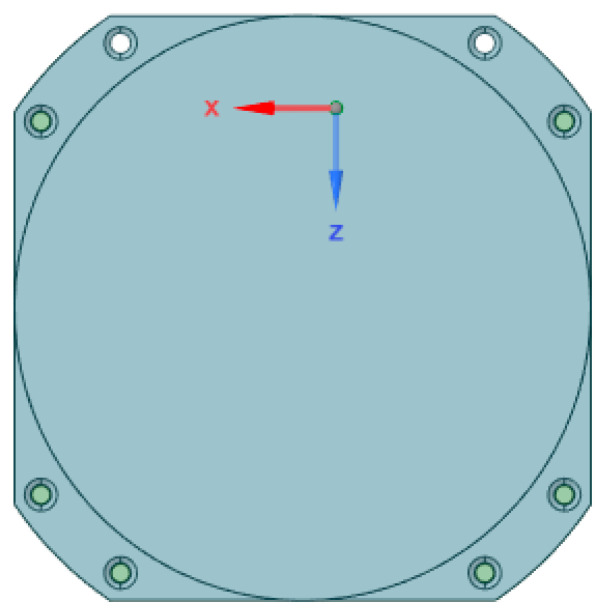
All-metal circular patch antenna [18].

**Figure 9 sensors-23-09832-f009:**
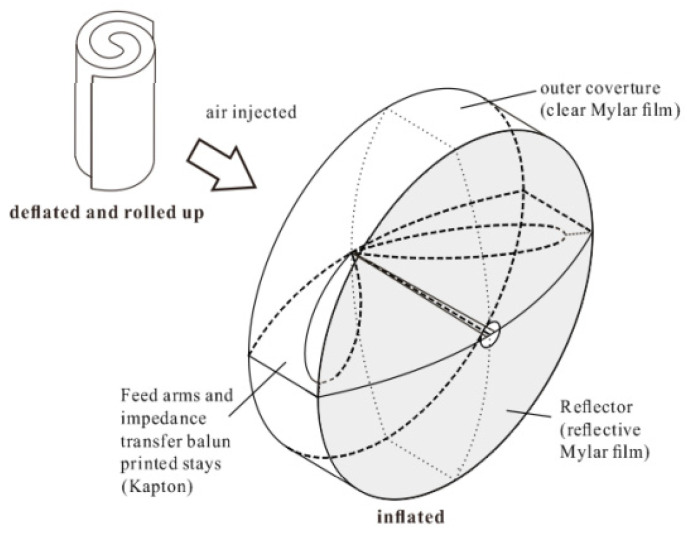
Inf-IRA concept and overall diagram [57].

**Figure 10 sensors-23-09832-f010:**
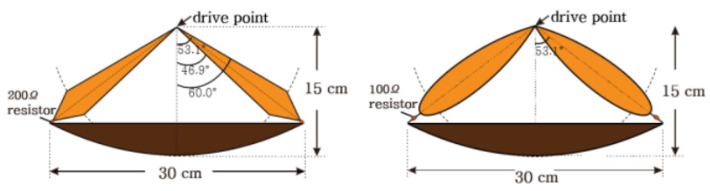
Side view of two antennas: conventional impulse-radiating antenna and asymptotic conical dipole fed impulse-radiating antenna [57].

**Figure 11 sensors-23-09832-f011:**
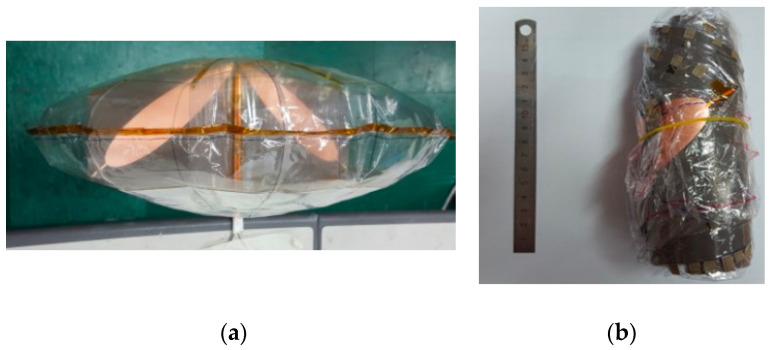
(**a**) Inflated and (**b**) deflated shape of fabricated inflatable impulse-radiating antennas [57].

**Figure 12 sensors-23-09832-f012:**
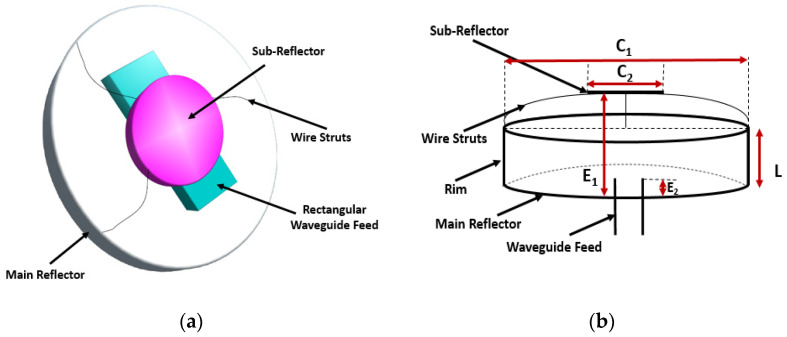
(**a**) All-metal single-element short backfire. (**b**) Waveguide inserted into the main reflector [60].

**Figure 13 sensors-23-09832-f013:**
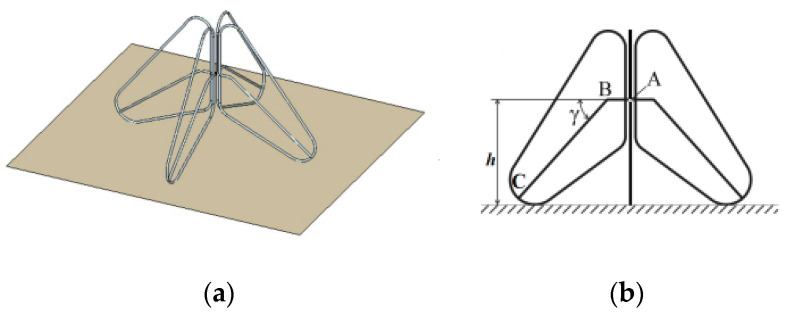
Small-sized active dipole antenna: (**a**) prototype, (**b**) antenna frontal projection. Adopted from “The first detection of the solar U+ III association with an antenna prototype for the future lunar observatory” by L. Stanislavsky et al. Research in Astronomy and Astrophysics, (2021) [71]. Reproduced by permission of RAA. All rights reserved.

**Figure 14 sensors-23-09832-f014:**
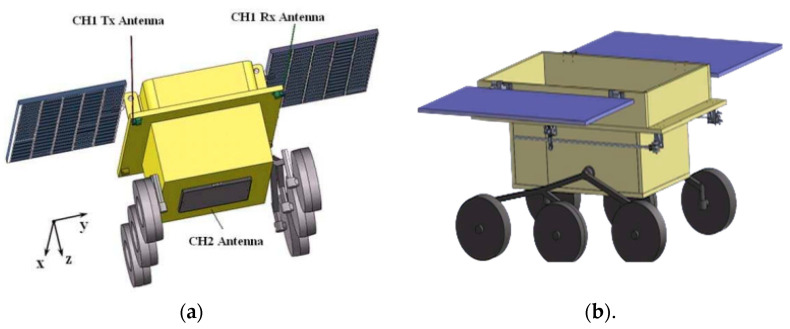
(**a**) The location of the LPR antennas on the lunar rover. (**b**) The folding of the CH1 antennas on the lunar rover. Adopted from Lunar Penetrating Radar onboard the Chang’e-3 mission by G.Y. Fang et al., Research in astronomy and astrophysics (2014) [75]. Reproduced by permission of RAA. All rights reserved.

**Figure 15 sensors-23-09832-f015:**
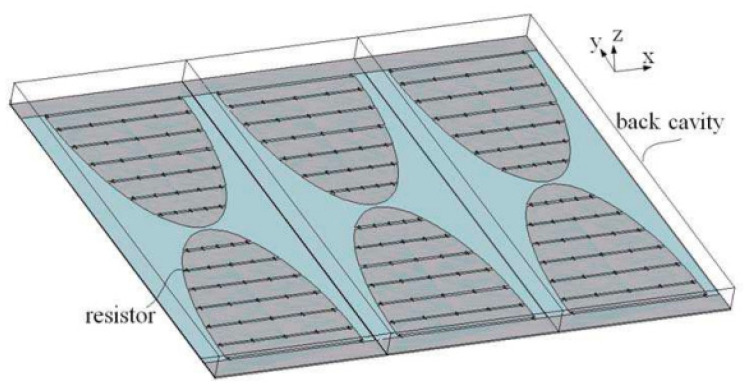
The structure of the CH2 antenna. Adopted from Lunar Penetrating Radar onboard the Chang’e-3 mission by G.Y. Fang et al., Research in astronomy and astrophysics (2014) [75]. Reproduced by permission of RAA. All rights reserved.

**Figure 16 sensors-23-09832-f016:**
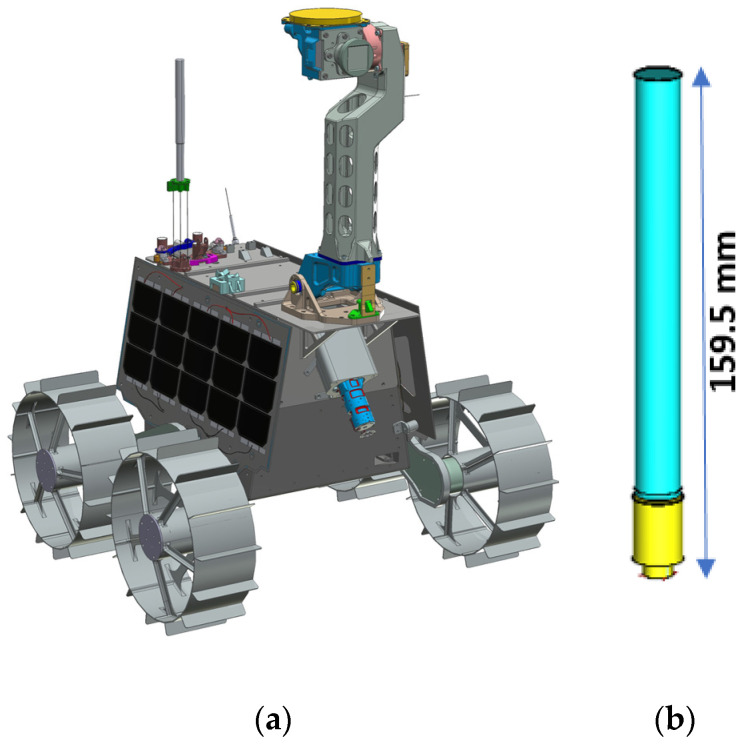
(**a**) Rashid rover with sleeve dipole antenna. (**b**) Proposed sleeve dipole structure. Proc. 2023 IEEE CAMA, Genoa, Italy, 2023 [17].

**Table 1 sensors-23-09832-t001:** Summary of lunar propagation models.

Ref	Models Considered						
	Free-Space	Freznel	Reflection	Two-Ray	Diffraction	Irregular Terrain	Multipath
[19]	✓	✓			✓	✓	
[20]					✓	✓	
[21]	✓			✓	✓		
[22]	✓			✓	✓		
[24]	✓					✓	✓

**Table 2 sensors-23-09832-t002:** QESA structural parameters.

Parameter	Value (mm)
w_sub_ and l	100
W	60.8
l_slot_	40
w_slot_	25
D	14
l_throat_	4.27
r_stub_	15
l_1_	1.7

**Table 3 sensors-23-09832-t003:** Comparison between planar lunar antennas in terms of application.

Antenna Type	Antenna Name	Ref	Frequency Band	Mission/Research	Application
Planar	CMMA (patch)	[44]	S band	Research	Omnidirectional antenna
	fHCFA (patch)	[44]	S and Ku bands	Research	Lunar surface network communications and planetary exploratory missions
	Vivaldi (array, slot)	[50]	L and S bands	Mission	Lunar regolith penetrating radar
	All-Metal Circular Patch Antenna	[18]	S band	Mission	Moon–Earth direct communication

**Table 4 sensors-23-09832-t004:** Comparison between planar lunar antennas in terms of performance.

Ref	Operating Frequency (MHz)	Bandwidth Fractional (%)	Gain (dBi)	S_11_ (dB at Centre Frequency)
[44]	2050	6.3	n/a	−24
[44]	2300 and 16,800	0.5 and 3	n/a	−29 and −27
[50]	1000–4750	130.43%	2–8	−15–(−17.5)
[18]	2101.2 and 2266	0.004	6	˂−15 dB

**Table 5 sensors-23-09832-t005:** Comparison between planar lunar antennas in terms of simulation parameters.

Ref	Size mm (W × L × H)	Substrate	e_r_	Software
[44]	12 × 12 × 11	Duroid 5880	2.2	Zeland’s IE3D electromagnetic simulator
[44]	5 × 5 × 4.5	Duroid 5880	2.2	Zeland’s IE3D electromagnetic simulator
[50]	100 × 100 × 1 (single antenna)	Polyimide	3.8	n/a
[18]	65 (diameter of circular patch)	Aluminium	-	FEKO

**Table 6 sensors-23-09832-t006:** All-metal single-element short backfire antenna dimensions.

Parameter	Value
C_1_	2λ_o_
C_2_	0.6λ_o_–0.7λ_o_
L	0.5λ_o_
E_1_	0.6λ_o_
E_2_	0.25λ_o_
λ_o_	35.61 mm

**Table 7 sensors-23-09832-t007:** Comparison between reflector lunar antennas in terms of application.

Antenna Type	Antenna Name	Ref	Frequency Band	Mission/Research	Application
Reflector	Inflatable Impulse-Radiating Antenna (inf-IRA)	[57]	VHF-S	Research	Radar
	Backfire (all-metal)	[60]	X-band	Research	Communication and navigation

**Table 8 sensors-23-09832-t008:** Comparison between reflector lunar antennas in terms of performance.

Ref.	Operating Frequency (MHz)	Bandwidth Fractional (%)	Gain (dBi)
[57]	50–4000	195.06	≅2–17
[60]	8920–9400 (cylindrical) 8000–8420 (hexagonal)	5.24 (cylindrical) 5.11 (hexagonal)	13.4–15.6 (cylindrical) 12.5–14.2 (hexagonal)

**Table 9 sensors-23-09832-t009:** Comparison between reflector lunar antennas in terms of simulation parameters.

Ref.	Size (Diameter in m)	Dielectric Materials Used (If Any)	e_r_	Software
[57]	0.3	Mylar and Kapton	2.8–3.7	FEKO (http://www.feko.info/, accessed on 4 October 2023)
[60]	0.07	n/a	n/a	Ansys HFSS

**Table 10 sensors-23-09832-t010:** Comparison between lunar wire antennas in terms of application.

Antenna Type	Antenna Name	Ref	Frequency Band	Mission/Research	Application
Wire	Dipole	[69]	HF	Mission	Near-side low radio frequency imaging
	Small-Sized Active Dipoles	[71]	HF	Research	Lunar radio telescopes
	CH1 (monopole)	[75]	VHF	Mission	Lunar penetrating radar
	CH2 (bow-tie)	[75]	VHF-UHF	Mission	Lunar penetrating radar
	Sleeve Antenna (dipole)	[17]	S	Mission	Lunar surface communication

**Table 11 sensors-23-09832-t011:** Comparison between lunar wire antennas in terms of performance and simulation parameters.

Ref	Operating Frequency (MHz)	Bandwidth Fractional (%)	Gain (dBi)	Size (mm)	Software
[69]	1–10	163.3	n\a	8 × 30.51	CST 2023
[71]	1–70	194.36	n\a	2.8 × 23	n/a
[75]	40–80	66.66	n\a	n/a	FEKO
[75]	250–750	100	−7.5	336 × 120	FEKO
[17]	2400–2700	17.64	2.23	12 × 159.5	CST

**Table 12 sensors-23-09832-t012:** List of recommended frequencies used in lunar communication and their applications.

Band	Frequency (GHz)	Application
UHF Band	0.3–1	Communication with lunar rovers and surface instruments Good penetration through lunar soil—ideal for surface operations
S	2–4	Telemetry and command functions in lunar missions Good balance between data rate and signal penetration through lunar regolith Suitable for tasks such as sending commands to spacecraft and receiving telemetry data
X	8–12	Higher data rate communication and scientific data transmission in lunar missions Transmission of larger data volumes, including high-resolution images and scientific data
Ka	26.5–40	Even higher data rates for high-data-rate science missions Particularly beneficial for missions with advanced instruments, including high-definition cameras and remote sensing equipment
G	110–300	Optical communication utilizes lasers for high-speed data transmission Potential for high-speed data transmission to and from lunar spacecraft Enables real-time communication for specific applications

## Data Availability

Data are contained within the article.

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
