# Peer review of "A Review of Lunar Communications and Antennas: Assessing Performance in the Context of Propagation and Radiation"

_sensors, 2023, doi:10.3390/s23249832_

Round 1
Reviewer 1 Report
Comments and Suggestions for Authors
Thank you for the contribution, well organised and full of useful informations.
Author Response
We appreciate your time and effort in reviewing the article.
Reviewer 2 Report
Comments and Suggestions for Authors
This is an interesting study to review the antenna structures for lunar communications. The comments for authors are as follows.
1. In the introduction section, the lunar environment was explained; however, it is not clearly mentioned how it can impact communication. Please mention the challenges specifically.
2. Also, it is recommended to add more explanation about the necessity of lunar communication and the types of required communication in the introduction section.
3. In page 6, paragraph 3, it is mentioned that there is an impact of the lunar environment on wireless communication. Please explain how the mentioned factors in this section impact the quality of wireless communication. A model or some results, e.g., SNR plots, are required to be included to support the statement.
4. It is appreciated that several antennas for lunar communication are included in this manuscript. It is recommended to mention why there are a variety of designs for each purpose and what are the limitations of each.
5. As it is mentioned in the introduction that the sleeve dipole antenna is the contribution of the authors, it is expected to have a more comprehensive section on this, including the design parameters and simulation and measurement results. Also, please elaborate further on the reason for selecting this antenna rather than other configurations.
6. How did the author simulate the lunar environment in the CST simulation? Please provide more details and include results that show the performance of the antenna in the lunar environment compared with standard simulation/measurement results.
7. It will be beneficial for the reader if more technical information is provided regarding the challenges in designing antennas for this application compared to terrestrial modeling.
Comments on the Quality of English LanguageThe manuscript needs careful proofread. Several errors, such as punctuation errors, can be seen in the current version.
Reviewer 3 Report
Comments and Suggestions for Authors
The manuscript "A Review of Lunar Communications and Antennas: Assessing Performance in the Context of Propagation and Radiation" focuses on lunar communication and antennas. It is interesting to researchers in related areas. Some improvements should be made.
1. Fig. 3 shows the planar, low-frequency, reflector, and wire antennas. The classification is not reasonable. Low-frequency antennas should not be listed as this. Please either classify them with frequency or with the types of antennas.
2. As a review paper, it is appreciated to summarize the communication frequencies used in the lunar, as well as their main functions.
3. The power capacity and communication data rate should be mentioned and discussed as well. Whether the antenna is a transmitting antenna or a receiving antenna? Is it for high-power or low-power applications? Please classify and summarize them.
4. A typical environment has to take the temperature and vacuum into consideration. Do the multipactor effects exist in the lunar for the design and applications of antennas?
5. The microwave power levels have been neglected in the review. Are there some large power antennas? It is interesting to discuss large power antenna applications in lunar.
6. S11 should be S_11.
7. In the conclusion part, a summary of the antenna applications in the lunar should be presented. It may be a guide for antenna design for the lunar communication applications.
Comments on the Quality of English LanguageThe English writing is OK.
Round 2
Reviewer 3 Report
Comments and Suggestions for Authors
Well revised.